# Exploration of the Noncoding Genome for Human-Specific Therapeutic Targets—Recent Insights at Molecular and Cellular Level

**DOI:** 10.3390/cells12222660

**Published:** 2023-11-20

**Authors:** Wolfgang Poller, Susmita Sahoo, Roger Hajjar, Ulf Landmesser, Anna M. Krichevsky

**Affiliations:** 1Department for Cardiology, Angiology and Intensive Care Medicine, Deutsches Herzzentrum Charité (DHZC), Charité-Universitätsmedizin Berlin, 12200 Berlin, Germany; ulf.landmesser@charite.de; 2Berlin-Brandenburg Center for Regenerative Therapies (BCRT), Charité-Universitätsmedizin Berlin, Corporate Member of Freie Universität Berlin and Humboldt-Universität zu Berlin, 13353 Berlin, Germany; 3German Center for Cardiovascular Research (DZHK), Site Berlin, 10785 Berlin, Germany; 4Cardiovascular Research Institute, Icahn School of Medicine at Mount Sinai, One Gustave L. Levy Place, Box 1030, New York, NY 10029, USA; susmita.sahoo@mssm.edu; 5Gene & Cell Therapy Institute, Mass General Brigham, 65 Landsdowne St, Suite 143, Cambridge, MA 02139, USA; rhajjar@partners.org; 6Berlin Institute of Health, Charité-Universitätsmedizin Berlin, 10117 Berlin, Germany; 7Department of Neurology, Brigham and Women’s Hospital, Harvard Medical School, Boston, MA 02115, USA; akrichevsky@bwh.harvard.edu

**Keywords:** immunology, innate immunity, immunogenetics, neurobiology, neurogenetics, noncoding genome, primate evolution genetics

## Abstract

While it is well known that 98–99% of the human genome does not encode proteins, but are nevertheless transcriptionally active and give rise to a broad spectrum of noncoding RNAs [ncRNAs] with complex regulatory and structural functions, specific functions have so far been assigned to only a tiny fraction of all known transcripts. On the other hand, the striking observation of an overwhelmingly growing fraction of ncRNAs, in contrast to an only modest increase in the number of protein-coding genes, during evolution from simple organisms to humans, strongly suggests critical but so far essentially unexplored roles of the noncoding genome for human health and disease pathogenesis. Research into the vast realm of the noncoding genome during the past decades thus lead to a profoundly enhanced appreciation of the multi-level complexity of the human genome. Here, we address a few of the many huge remaining knowledge gaps and consider some newly emerging questions and concepts of research. We attempt to provide an up-to-date assessment of recent insights obtained by molecular and cell biological methods, and by the application of systems biology approaches. Specifically, we discuss current data regarding two topics of high current interest: (1) By which mechanisms could evolutionary recent ncRNAs with critical regulatory functions in a broad spectrum of cell types (neural, immune, cardiovascular) constitute novel therapeutic targets in human diseases? (2) Since noncoding genome evolution is causally linked to brain evolution, and given the profound interactions between brain and immune system, could human-specific brain-expressed ncRNAs play a direct or indirect (immune-mediated) role in human diseases? Synergistic with remarkable recent progress regarding delivery, efficacy, and safety of nucleic acid-based therapies, the ongoing large-scale exploration of the noncoding genome for human-specific therapeutic targets is encouraging to proceed with the development and clinical evaluation of novel therapeutic pathways suggested by these research fields.

## 1. Introduction

While it is well known that 98–99% of the human genome does not encode proteins, but are nevertheless transcriptionally active and give rise to a broad spectrum of noncoding RNAs (ncRNAs) with complex regulatory and structural functions, specific functions have so far been assigned to only a tiny fraction of all known transcripts [1,2,3,4,5,6,7,8,9]. While massive expansion of the noncoding genome relative to the protein-coding sequences during evolution from simple to complex organisms is obvious, *functionality* of transcription of almost the *entire* genome is not. There will be transcriptional noise, and involvement of transcription of repeats, pseudogenes, retroviruses, and so on. The fraction of functional noncoding transcripts across the huge spectrum of highly diverse species must certainly await final determination in a more remote future.

The critical, but so far essentially unexplored, impact of the noncoding genome in human health and disease [8,10,11,12] is nevertheless strongly suggested by the overwhelmingly expanding fraction of ncRNAs, sharply contrasting with only a modest increase in the number of protein-coding genes, during evolution from simple organisms to humans. Importantly, deep research into the vast realm of the noncoding genome during the past decades has led to a fundamentally enhanced appreciation of the multi-level complexity of the human genome.

Here, we address a few of the many remaining knowledge gaps, and consider some newly emerging topics and strategies of research. We provide a critical up-to-date assessment of recent insights obtained by molecular and cell biological methods and by application of systems biology approaches, focusing upon two topics of high current interest. First, since several evolutionary recent noncoding regions of the human genome have been discovered to exert critical regulatory functions in a broad spectrum of cell types (neural, immune, cardiovascular), might these regions constitute novel therapeutic targets in human diseases? Second, noncoding genome evolution has been shown to be causally linked to brain evolution, and increasingly profound interactions between mind and brain and the immune system in humans are emerging. Thus, how could human-specific brain-expressed genes play a direct or indirect (immune-mediated) role in human diseases?

“Human-specific” in this context does not necessarily mean there are no paralogues or similar genes in other species, but that the respective gene has gained novel or particularly important functions in humans. This may simply occur because an organ such as the brain has dramatically increased in size and complexity during evolution, requiring more sophisticated genome-level regulation and spatiotemporal coordination of cellular functions and cell–cell interactions.

Caution should still be exercised when assuming that humans are the most complex organism from a molecular genetic perspective, too. Maybe the overwhelming number of ncRNAs found in humans is, at least in part, a consequence of humans being the most extensively studied organism. Currently, it cannot be excluded that equally deep investigation of more “primitive“ species, e.g., octopuses or amoebae, would show an unexpectedly high proportion of ncRNAs. Future research, covering a more comprehensive spectrum of species, may unveil unexpected new insights regarding the role of the noncoding genome during evolution in general.

## 2. Overwhelming Expansion of the Noncoding Genome in Higher Organisms

Decades ago already, landmark studies documented the intracellular presence of large amounts of RNAs that are transcribed, but do not encode proteins. Only part of these ncRNAs could later be linked to mRNA splicing (e.g., small nuclear RNAs [snRNA]), or were specifically involved in the translation machinery and its regulation (e.g., transfer RNAs [tRNAs], ribosomal RNAs [rRNAs], small nucleolar RNAs [snoRNA], but the vast majority remained functionally cryptic for a long time. More recent studies then led to the discovery of entire new classes of small RNAs, generated not only by novel biosynthetic pathways and mediating gene expression post-transcriptionally (e.g., microRNA [miRNAs]), but also unusually large long ncRNAs [lncRNAs], in almost all cases, of still unknown functional significance. Broad evidence derived from multiple genetic, biochemical, and other experimental and clinical studies during the past decades clearly revealed a key role of ncRNAs in the genetic programming of complex organisms, during their development and in health and disease.

The steeply increasing fraction of ncRNAs in the genome during evolution, from simple to highly complex organisms, strongly contrast with the only modest increase in the number of protein-coding genes (Figure 1) and is consistent with the assumption of an overwhelming role of the ncRNA species in higher organisms. Both *Caenorhabditis elegans* with ~1000 somatic cells and humans with ~30 × 10^12^ somatic cells have ~20,000 protein-coding genes (‘g-value paradox’) [13]. Regarding disease pathogenesis, it is evident since the Encyclopedia of DNA Elements (ENCODE) project [3] that confinement to the analysis of protein-coding regions of the human genome is inadequate, because many noncoding variants are associated with important human diseases. Inclusion of the noncoding genomic elements in pathogenetic studies seems mandatory and one approach is comprehensive transcriptome mapping encompassing protein-coding genes, as well as diverse small and large ncRNAs.

Apparently, ~5% of the human genome are functionally constrained [14]. Since only ~1.5% of the genome could possibly be assigned to protein-encoding genes, a major part of these constraints is necessarily associated with functionally important conserved noncoding elements (CNEs), preserved among organism through millions of years of evolution at the cellular and systemic level. As one example, there is evidence for higher-order genome organization functions of lncRNAs in diverse cell types and cell lineages, and during cell differentiation [15]. It was recently found that not only transcription, but also translation, is pervasive outside of protein-coding regions (e.g., lncRNAs, 3′-untranslated regions, introns). Although resulting polypeptides are generally nonfunctional, their translation is considered relevant for the emergence of novel functional genes [16].

One particularly interesting but highly complex group of ncRNAs are those designated long noncoding RNAs (lncRNAs). The term ‘lncRNAs’ encompasses RNA polymerase I (Pol I), Pol II, and Pol III transcribed RNAs, and RNAs from processed introns that are at least 200 nt in length. More than 100,000 human lncRNAs have been identified, many of which are primate-specific [17,18,19,20,21,22,23,24]. A recent consensus statement addressed definition, nomenclature, conservation, expression, phenotypic visibility, structure, and functions of lncRNAs [25]. The paper emphasizes that many lncRNAs are cell lineage-specific, associated with developmental enhancers, and likely contribute to species diversity and evolution. Segments of lncRNAs may maintain sequence conservation comparable to protein-coding genes, exhibit conserved exon-intron structures and splice junctions, and retain orthologous functions despite rapid sequence evolution [26,27,28,29].

Importantly, noncoding genome research during the past decades not only revealed unanticipated multi-level complexity of the genome in higher organisms, but has also inspired fundamentally new therapeutic avenues as outlined in Figure 2.

## 3. Recent Searches for Evolutionary Relevant Primate/Human-Specific Noncoding Genes

*Landmark studies*: Nobel-prize winning work by Pääbo et al. [30], based on experimentally revolutionary work ultimately enabling deep sequencing of the genomes of extinct primates and hominids, has revolutionized a field of research which was previously the exclusive realm of archaeologists and paleoanthropologists. The puzzle to mankind, when and where modern humans originated and how they differ from and interacted with other now-extinct forms of humans is now being addressed by molecular geneticists [30,31,32,33,34,35,36,37,38,39] and protein biochemists [40] also. These recent developments have enabled revolutionary novel approaches to diverse and so far enigmatic issues such as the evolutionary development of speech [41].

Screening for evolutionarily relevant genetic changes during the diversion of humans from the other primate species first focused on protein-coding genes with a favorable signal/noise ratio and commonly more obvious functional impact, as assessed by conventional molecular and cell biological approaches. Almost two decades ago, however, after the discovery that 98.5% of the human genome is transcribed but not translated into proteins, a first study scanned for “human accelerated regions“ [HARs] with accelerated substitution rates in the human lineage [42]. This pioneering work identified one region (HAR1) expressed during human cortical development as the most significantly altered element. While the two transcripts from this region (HAR1F and HAR1R RNAs) lacked protein-coding potential, HAR1F transcript folded into a stable RNA structure and occurs with ncRNAs such as miRNAs and many lncRNAs [42]. Since the first study [42], further HARs were identified and investigated; as one important discovery, their proximity to neuropsychiatric disease genes was revealed, which is discussed in more detail below.

*Brain transcriptome*: Pembroke et al. [43] recently discussed the evolutionary conservation and divergence of the human brain transcriptome in general. They emphasize that, although mouse models allow dissection of genetic effects on molecular, cellular, physiological, and behavioral brain phenotypes, the extent to which neurological or psychiatric traits are human- or primate-specific, and cannot be faithfully recapitulated in mouse models, is unknown. Pembroke et al. name multiple human neuropsychiatric and neurodegenerative disease risk genes (COMT, PSEN-1, LRRK2, SHANK3, SNCA) with grossly divergent expression pattern in mice vs. humans, for some of which functions at the cellular level have recently been assigned. We review similar and possibly critical knowledge gaps in the cardiovascular field, which are likewise due to differences between the noncoding genome in humans vs. experimental animals (Section 6).

Espinos et al. [44] discuss in-depth genetic mechanisms regulating the evolution of cortical neurogenesis. They address genes which emerged in the recent human and primate lineages and apparently promote cortical progenitor proliferation and increase neurogenesis. These include structurally conserved primate lncRNAs transiently expressed during human cortical differentiation and modulating the expression of cell type-specific genes sequentially activated during cortical neurogenesis [45]. Several HARs [46,47] involved therein are predicted to act as regulatory enhancers [48] and are located in the vicinity of genes important for brain development [49].

## 4. Exploration of the Noncoding Genome for Human-Specific Therapeutic Targets—Three Levels of Increasingly Complex Genome–Disease Relationships

Obviously, any potential pathogenic role of “human-specific” protein-coding genes, as well as of noncoding RNAs, will remain hidden during studies of animal models (murine and diverse others) of human diseases. In principle, any such gene may play an important role in some human disease, however.

It is important to note that “human-specific” in this context does not necessarily mean there are no paralogues in other species, but that the respective gene has gained novel or particularly important functions in humans. This may occur, for instance, because an organ such as the brain has dramatically increased in size and complexity during evolution. Such genes are more likely to play a pathogenic role in diseases (e.g., neuropsychiatric), which are relatively unique to or display distinct phenotypes in humans.

From a technical standpoint, it is essential to exercise caution when designating a lncRNA as specific to humans or primates. This is because in numerous species, the non-protein coding regions are frequently incompletely annotated, and lncRNA databases remain inadequately established. Additionally, since many lncRNAs are expected to play roles in local chromatine remodeling (i.e., function in-cis), not only *sequence* conservation, but also *positional* conservation in the context of genomic location should be considered.

At a first level, human-specific proteins could *directly* alter known signal pathways or other canonical cellular functions. One example is interleukin 8 (IL8), which emerged as one of the most strongly deregulated genes in peripheral blood mononuclear cells (PBMCs) after human myocardial infarction [8,10,50] but does not exist in mice, the most used animal model in cardiovascular research. Drury et al. [51], investigating the evolution and emergence of interferon regulatory factor 9 (IRF9), a key component of the ISGF3 complex and the cellular innate immune response, identified primate-specific IRF9 (PS-IRF9) isoforms unique to old world monkeys and great apes. Ellwanger et al. [52] analyzed the function of the primate-specific NLRP11 gene product and found it highly expressed in human immune cells (myeloid cells, B cells, lymphoma lines). This identified a novel role in the regulation of inflammatory responses in humans.

At a second level, human-specific noncoding RNAs non-existent in animal models may *directly* alter cellular functions and cell–cell interactions through non-canonical mechanism, and via complex interactions with other proteins and ncRNAs. An interesting recent study found a primate-specific lncRNA (CHROM***R***) to be induced by SARS-CoV-2 infection coordinated expression of interferon-stimulated genes (ISGs), and restricted viral infection of macrophages [53]. Another study [54] found the primate-specific lncRNA (CHROM***E***) elevated in plasma and atherosclerotic plaques of coronary artery disease (CAD) patients. Gain- and loss-of-function approaches showed that CHROM***E*** promotes cholesterol efflux and HDL biogenesis and constitutes a central component of the non-coding RNA circuitry controlling cholesterol homeostasis in humans. Both CHROM***R*** and CHROM***E*** are antisense RNAs.

Neuroscience has likewise identified brain-expressed lncRNA several of which are primate/human-specific and associated with brain development and neuropsychiatric diseases, as outlined in Section 7. Any of these could *directly* induce damage at their sites of expression in the brain.

At a third level, *indirect* impact of human-specific genes and ncRNAs upon pathogenesis may be exerted via peculiar interactions between human brain and mind and the immune system. Interactions between the immune system and the nervous system were initially described in the context of diseases. More recent studies have begun to reveal how immune cell-derived effectors can influence host behavior even in the absence of infection [55]. Essentially, the immune system shapes nervous system function and controls manifestations of host behavior. In the context of evolution, interactions between these two highly complex biological systems may have evolved to maximize an organism’s ability to respond to environmental threats in order to survive [56].

## 5. Genetics of Immune System and Neuro-Immune Interactions Impact upon a Large Spectrum of Human Diseases

From the discovery that about 98–99% of the human genome do not encode proteins, but instead generate a broad spectrum of ncRNAs many of whom are involved in the immune response [57,58,59,60,61,62,63,64,65,66,67,68,69,70,71,72,73,74,75,76,77,78,79,80,81,82,83,84,85,86,87], decades passed until finally successful clinical exploitation of ncRNAs and of novel drugs developed using them as blueprints was achieved [8,10] (Figure 2). Across the entire spectrum of medical disciplines, it has been ascertained that the non-coding genome plays a key role in genetic programming and gene regulation during development as well as in health and disease (Figure 3 and Figure 4).

A particularly important biological network critically involved in multiple human diseases is the immune system. It is paramount to understanding disease pathogenesis, and to open new therapeutic avenues, from cardiovascular medicine to neurology and other clinical disciplines. For instance, in-depth molecular and genetic analyses of innate immunity have led to the identification of novel molecular players and therapeutic targets in cardiovascular diseases [89]. Immunity in general is deeply involved in many processes which are discussed more specifically in the following chapters. In this context Silverstein [56] has thoughtfully suggested that, in contrast to ”Darwinian” evolution involving adaptation to past challenges, evolution has “devised” two unique biological mechanisms permitting to anticipate future challenges: adaptive immune response and neural memory functions.

The study of immune systems evolution revealed differences, but also striking similarities of the immune mechanisms across different taxa in the context of evolution [90,91]. Major impact of the noncoding genome upon functions and stability of the immune response against diverse challenges has long been appreciated. There is a broad spectrum of broadly diverse non-coding RNAs (ncRNAs) involved in the human immune response [57,58,59,60,61,62,63,64,65,66,67,68,69,70,71,72,73,74,75,76,77,78,79,80,81,82,83,84,85,86]. Some of these have been identified as primate-specific, e.g., the lncRNA CHROM***R*** [53]. These and related discoveries highlight diverse clinically relevant peculiarities of human immunology beyond the inbred mouse model [92].

## 6. Noncoding Genomic Regions Impact upon Cardiovascular Pathogenesis in Humans

Within the field of cardiovascular medicine, a number of early experimental studies [93,94,95,96,97,98,99,100,101,102] revealed that certain ncRNAs (miRNAs) are regulators of cardiovascular homeostasis in animal models. This of course immediately suggested they might have potential to improve diagnostics and could possibly even be developed into novel therapeutics. The road to in-depth understanding of the molecular workings of at least a few of the numerous ncRNA classes, however, and beyond that the development of highly sophisticated bioengineered nucleic acid drugs [93,103,104,105,106,107,108,109,110,111,112,113] (Figure 2, Figure 5 and Figure 6) which are critically required to render them safe and efficacious for clinical applications, required critical input from several disciplines and two decades, counting from the early experimental work to the first clinically successful trials. In Section 12 below, we update the conceptual and methodological challenges on the road towards clinical exploitation of potential novel human-specific therapeutic targets.

Particularly advanced is the development of RNA interference [RNAi] drugs, which use recently discovered pathways of endogenous short interfering RNAs (siRNAs) and have become highly versatile tools for the efficient silencing of any protein-coding or noncoding transcript and gene. A series of ground-breaking clinical trials [108,112,113,114,115,116,117,118,119,120,121,122,123,124,125,126,127,128,129,130,131,132,133,134,135] has provided definite evidence of therapeutic potential of RNA interference [RNAi] and antisense drugs for cardiovascular disorders. For more in-depth coverage of the enormously challenging bioengineering, safety, and regulatory hurdles to be overcome towards clinical therapy during the past decades, we may refer the reader to comprehensive recent reviews [8,10,136].

As examples of human-specific noncoding RNAs non-existent in animal models, we invoke two primate-specific lncRNAs. One of these (CHROM***R***) [53] is induced by SARS-CoV-2 infection, coordinates expression of interferon-stimulated genes (ISGs) and restricts viral infection of macrophages [53]. Given the immense and diverse impact of SARS-CoV-2 upon human health, existence of this primate-specific lncRNA is of interest per se. Moreover, the human host’s innate immune response plays a critical role in COVID-19 pathogenesis [137,138,139,140]. Thus, several SARS-CoV-2 accessory proteins have been suggested to affect the innate immune response. Abnormal pro-inflammatory cytokine levels and immune cell infiltration are associated with tissue damage severity and morbidity. Overall, dysregulation of the host immune response and elevated cytokine release are crucial factors for the severity of COVID-19, and thus any peculiar human-specific immunoregulatory ncRNA deserves attention.

Another primate-specific lncRNA is CHROM***E*** [54]. It is elevated in plasma and atherosclerotic plaques of coronary artery disease (CAD) patients. Gain- and loss-of-function experiments revealed that CHROM***E*** promotes cholesterol efflux and HDL biogenesis and constitutes a key component of the ncRNA circuitry controlling cholesterol homeostasis in humans. Both publications illustrate how, after decades of traditional research in the respective fields, investigation of the noncoding genome still uncovers unexpected novel molecular players. These may significantly contribute to improved pathogenic understanding, beyond what could possibly be derived from animal models. Similar to these examples from the cardiovascular field, studies from the neurosciences and clinical neurology identified impact of species-specific transcripts upon human brain development and neurological disorders [45,141,142,143,144,145,146].

## 7. Primate/Human-Specific ncRNAs in Neural/Neuroimmune Cells and Their Impact upon Brain Development and Neuropsychiatric Disorders

There is broad and growing interest in the identification of primate-specific genes involved in primate evolution and the evolution of humans. This includes not only coding genes, but also diverse types of noncoding transcripts, e.g., miRNAs [141] and lncRNAs [147]. One driver of this interest, beyond deeper understanding of human evolution [30], is a hope [141] to identify unique features of human brain development and function which may be critical towards the elucidation of higher cognitive functions, and of human-specific pathologies like neuropsychiatric and behavioral disorders.

After a first study had identified and partially characterized a “human accelerated region” (HAR) [42], further HARs were investigated and indeed, as one important discovery, their proximity to neuropsychiatric disease genes was revealed [39,148,149,150,151,152,153,154]. Since many HAR-associated genes are regulators and hubs in transcriptional networks, their differential expression would affect many other genes and cellular processes, suggesting outsized effects caused by noncoding HAR mutations. HAR enhancers may help to discover the genetic basis for disease, while medical genetics may reveal which HAR variants are pathogenic.

Further along this line, drug target data were employed to map neuropsychiatric disorders to HARs via nearby genes [155,156,157,158]. These genetic studies, involving large neuropsychiatric patient populations, suggest that human-specific noncoding transcripts from HARs as well as species-specific miRNAs are involved in human-specific pathologies like psychiatric and behavioral disorders. If so, sequence variants or altered expression pattern of these ncRNAs *in the brain* would *directly* lead to brain dysfunction and disease.

Conveniently, more than two decades of research into the development of diverse types of RNA therapeutics towards clinical applications (as discussed above for cardiovascular and metabolic diseases) has set the stage for targeting/silencing of essentially any type of transcript considered relevant in other disease fields, too. In addition to RNA interference (RNAi) drugs, antisense therapeutics and CRISPR/Cas9-based approaches are also being evaluated with regard to their potential for the treatment of neuropsychiatric disorders [159,160]. Since many ncRNAs are specifically enriched in the central nervous system [CNS], and their dysregulation implicated in Alzheimer’s disease and related dementias, Nguyen et al. [161] review conventional small molecule drugs targeting ncRNA as possible therapeutics for Alzheimer’s disease and related dementias (ADRD).

Noncoding RNA targets: lncRNAs play important roles during normal brain development and in the pathogenesis of neurodegenerative disorders (ND) [162]. One target of particular therapeutic interest is *Nuclear enriched abundant transcript 1* (*NEAT1*) (Figure 3), which plays a role in mediating nuclear retention of TAR DNA-binding protein 43 (TDP-43) and is potentially protective in certain proteinopathies [163,164,165,166,167,168] involving aberrant protein aggregates comprising tau, amyloid-β, and α-synuclein. TDP-43 is a highly conserved nuclear RNA/DNA-binding protein regulating RNA processing. Accumulation of TDP-43 aggregates in brain is common to ND such as amyotrophic lateral sclerosis, frontotemporal dementia, and Alzheimer’s disease (AD). Relevant with regard to possible therapeutic options, *NEAT1* is upregulated in AD temporal cortex and hippocampus. Hippocampal knockdown of *NEAT1* with siRNA improved memory in aged mice and vice versa [169]. Viral knockdown of *NEAT1* rescued memory deficit in APP/PS1 mice [170].

Further lncRNA targets of interest include activity-dependent transported *Adeptr* [171], brain cytoplasmic BC200 [172,173,174,175], BACE1-AS (antisense relative to β-secretase 1 gene) [176,177,178], and *Carip* [179]. In addition, other types of ncRNAs (miRNAs, piRNAs, lncRNAs, circRNAs) are dysregulated in AD and related dementias with first evidence for therapeutic potential [161] (Figure 7). Thus, miR-132 is downregulated in AD hippocampus, prefrontal, and temporal cortex. Viral overexpression as well as miR mimics rescued hippocampal cell death, tau homeostasis, hippocampal adult neurogenesis, and behavioral deficits in various AD mouse models [180,181,182]. miR-195 is downregulated in AD parietal cortex. Viral overexpression in mouse models decreased Aβ plaque, tau hyperphosphorylation, and rescued cognitive deficits in ApoE4^+/+^ mice [183].

These first experimental therapeutic studies hold promise for further developments with translational potential employing highly sophisticated and safe drug delivery systems. These should have well-established molecular and cellular mechanisms of action, including carrier ligand–cell surface receptor [184,185,186,187,188,189,190] interactions, as well as clearly characterized *in vivo* behavior [191,192,193].

## 8. Multi-Level Functional Integration of Extended Noncoding Regions of the Human Genome—Critical Impact upon Fundamental Cellular Processes Governing Immune Response and Oncogenesis

To illustrate the multi-level functional integration of major regions of the human genome, above and beyond individual ncRNAs, we invoke two examples addressing genome-level immunoregulation (*NEAT1–MALAT1* cluster) (Figure 3), and neoplastic transformation in glioblastoma (HOXD-embedded HOXD-AS2 and distant enhancer-associated LINC01116) (Figure 4).

The evolutionary conserved *NEAT1–MALAT1* cluster encounters interest in cardiovascular medicine [50,194,195], oncology [196,197,198,199,200,201], and neurosciences and clinical neurology [170,202,203,204]. While single studies from these fields identified pathogenic roles in specific disease settings, broad interdisciplinary interest apparently results from a deep-rooted complex stabilizing function of the cluster. Within the cardiovascular field, suppression of lncRNA *NEAT1* was observed in circulating immune cells of post-myocardial infarction (MI) patients. Mice devoid of *NEAT1* or *MALAT1* displayed immune disturbances affecting monocyte-macrophage and T cell differentiation, and an immune system highly vulnerable to stress stimuli [205] and prone to the development of atherosclerosis. Uncontrolled inflammation is a key driver of multiple other diseases (see Section 7 below), too, which may underly the current broad interdisciplinary interest.

The structure, organizational levels and functions of the noncoding genome are still largely unexplored [8,10,25]. One aspect of particular interest with regard to function(s) is the common unusual complexity of lncRNA interactions with other ncRNAs, proteins, and cellular and subcellular membrane components, as well as their posttranscriptional processing and intracellular kinetics. As an example, the human *NEAT1–MALAT1* cluster generates lncRNA remaining nuclear, whereas tRNA-like transcripts (mascRNA, menRNA), enzymatically generated from these precursors, translocate to the cytosol. *NEAT1-/-* and *MALAT1-/-* mice display massive atherosclerosis and vascular inflammation [50,194,195,206]. A recent study found that these tRNA-like molecules are critical components of innate immunity and contribute to a balance response of immune cells to diverse challenges. They appear as prototypes of a new class of noncoding RNAs distinct from others (miRNAs, siRNAs) by biosynthetic pathway and intracellular kinetics.

For the long primary transcripts of *NEAT1* a particularly interesting function of general cell-biological interest has been elucidated in much detail. These lncRNAs are critical for the formation of paraspeckles which are involved in multiple cellular functions [205]. Yamazaki et al. have put the phenomenon of paraspeckle formation into a much broader context of micellization and the formation of biomolecular condensates [207], which are essential for subcellular and nuclear compartmentalization. Molecules involved in these fundamental processes may have deep impact upon various cellular functions in a context-dependent manner (e.g., immune stress, infections, toxins). Association of such molecules with diverse diseases is therefore not entirely unexpected.

Other recent studies [88,208,209] have elucidated–– in mechanistic detail–– the complex multi-level functional integration of distinct regions of the human genome, expressing lncRNAs, in the molecular pathogenesis of glioblastoma (Figure 4).

A remarkable molecular circuit involving ncRNAs plays a pivotal role in governing cell fate and transformation within the brain. This circuit comprises a miRNA, lncRNAs, and a small nuclear RNA (snRNA). The initial component of this intricate network, miR-10b, was originally identified as a unique miRNA that remains transcriptionally silenced in normal brain cells but becomes derepressed in low-grade gliomas and nearly all adult high-grade gliomas, including the most aggressive glioblastoma (GBM), a highly heterogeneous class of brain tumors [210,211]. While activated in glial lineage cells, miR-10b functions through both classical and non-conventional pathways. It regulates the expression of multiple mRNAs associated with cell cycle, cell death, and invasion, such as CDKN1A, CDKN2A, BIM, as well as ncRNA targets [210,212,213]. Notably, an unbiased analysis of miR-10b targets has identified an essential ncRNA component of the spliceosomal machinery, U6 snRNA, as a principal direct target. By modulating the structure, modifications, and levels of U6 snRNA, miR-10b exerts influence over the splicing of numerous cancer-related genes [213].

Overall, glioma cells display a strong dependence on miR-10b, making this small tu-mor-promoting RNA an attractive target for the development of GBM therapies. Various miRNA-targeting modalities, formulations, and delivery approaches employing antisense oligonucleotides (ASOs), gene editing, and small molecules, are currently under development [214,215,216,217].

The intriguing phenomenon of miR-10b’s silencing in neuroglial cells of the brain and its transcriptional activation during gliomagenesis has prompted investigations into the up-stream mechanisms responsible for this activation. Interestingly, the entire miR-10b locus, which also encodes 12 HOXD genes, becomes activated in gliomas through a mechanism involving lncRNA-mediated spatial chromatin reorganization (looping). Two interacting lncRNAs, HOXD-AS2 and LINC01116, associate with the HOXD3/HOXD4/miR-10b promoter and a distant enhancer, respectively, and both are necessary for the locus’s derepression and gene expression ([88] and Figure 4).

Furthermore, recent work suggests that the LINC01116 enhancer RNA (eRNA) also acts in trans and exhibits global genome wide-modifying activity. It directly binds to more than a thousand gene promoters, including those of 44 glioma-specific transcription factors distributed across all chromosomes, derepressing them by removing the Polycomb repressive complex 2 (PRC2). Consequently, the activation of this single lncRNA in astro-cytes, which are cells of glioma origin, is sufficient to trigger the glioma transcriptional program. In addition to miR-10b and HOXD factors, this includes the activation of neuro-developmental regulators like OLIG2, SOX2, POU3F2, and SALL2, along with multiple oncogenes such as EGFR, PDGFR, TERT, BRAF, and miR-21, ultimately leading to malignant transformation [208]. Conversely, targeting LINC01116 with siRNA or ASO may hold substantial therapeutic potential for malignant gliomas, opening exciting new avenues in neuro-oncology and, more broadly, neurology.

It is worth noting that this circuit encompasses various types of ncRNAs, both in terms of evolutionary conservation and functional diversity. These include the highly conserved and functionally essential in diverse eukaryotic species U6 snRNA, the relatively well-conserved miR-10b in vertebrates, and the largely primate-specific LINC01116. These observations prompt intriguing questions about the origin of human brain tumors, a phenomenon uncommon in other mammals, with only a few exceptions. This parallels the scenario seen in most human neurologic diseases.

The recent evolutionary emergence and the species- and cell-type specificity of LINC01116 and many other lncRNAs invite further in-depth research, undoubtedly lead-ing to better understanding of human pathologies and the discovery of new therapeutic targets and biomarkers for various diseases. Furthermore, with regard to LINC01116, it is noteworthy that this transcript is expressed at high levels not only in the brain, but also in some normal and cancer cells outside the central nervous system. Additional studies into its functions beyond the CNS are needed. It is also imperative to integrate our under-standing of its chromatin-modifying activity with potential extra-nuclear functions [218,219].

## 9. Impact of Human-Specific Genes and ncRNAs upon Pathogenesis via Peculiar Interactions between Human Brain and Mind and the Immune System

Considering three levels of increasingly complex genome–disease relationships as outlined in Section 4, at the third level peculiar *indirect* pathways from the noncoding genome to neurological diseases may result from the well-documented close and intense interactions between human brain and mind and the immune system. In the latter, immunoregulatory ncRNAs play important stabilizing roles and some of these, e.g., CHROMR, are primate-specific [53]. Sequence variants or altered expression pattern of these ncRNAs in immune cells may *directly* trigger brain dysfunction.

The grave adverse effect of psychological stress upon human diseases is well known [220,221,222], and several stress-induced brain–immune system interactions have been elucidated at the molecular, cellular, and systemic level [55,221,223]. In fact, the field of psychoneuroimmunology is one of the fastest-growing fields in the life sciences aiming to stepwise elucidate the highly complex interactions between nervous system and immune system at the molecular and cellular level [224,225,226,227,228,229].

Since psychological stress is certainly different in humans as compared to all other species, as a consequence of the peculiarity of the human brain and mind [158], any *direct* genetics-based immune dysfunction (Section 5) may well synergize with any *direct* genetic predisposition towards neuropsychiatric disorders (Section 7), *indirectly* resulting in grave brain dysfunction and disease once sufficient psychological stress peculiar to humans is imposed.

## 10. Current Status of Translational Research into Nucleic Acids-Based Therapeutics

Nucleic acid-based and nucleic acid-targeting therapeutics are currently developed at large scale for the prevention and management of multiple diseases for several reasons:

(1) Genetic studies have highlighted novel therapeutic RNA targets suggested to be causal for these diseases;

(2) There is substantial recent progress in delivery, efficacy, and safety of nucleic acid-based therapies;

(3) They enable effective modulation of therapeutic targets that cannot be sufficiently or optimally addressed using traditional protein-targeting small molecule drugs or antibodies.

Nucleic acid-based therapeutics in development also being evaluated for the diseases outlined and discussed above include: mRNA-targeting drugs for gene silencing; miRNA inhibitors and mimics; gene augmentation therapies; genome-editing approaches:

(A) *mRNA-targeting drugs for gene silencing*: several large-scale clinical development programs, using antisense oligonucleotides [ASO] or short-interfering RNA [siRNA] therapeutics for prevention and management of cardiovascular disease have been initiated. These include ASO and/or siRNA molecules to lower apolipoprotein (a), proprotein convertase subtilisin/kexin type 9 (PCSK9), apoCIII, ANGPTL3 for the prevention and treatment of patients with atherosclerotic cardiovascular disease. In other fields of medicine including neurology, silencing of transthyretin (TTR) was evaluated for the treatment of TTR amyloidosis [10,133,230,231,232].

(B) *miRNA mimics and inhibitors for miRNA modulation*: Several types of mimetic drugs and inhibitors (e.g., antagomirs) have been developed and a few of them evaluated in clinical trials [135]. These drugs may be “classical” small molecule drugs [233,234,235] or engineered nucleic acids [106,217].

(C) *Gene augmentation therapies*: EMA/FDA have approved “classical” gene therapies, i.e., those involving vector-based transfer of the protein-coding cDNA sequence into the patient, for monogenic disorders. These include the hemophilias [236,237,238,239,240,241,242], homozygous hypercholesterolemia [243], and others. Very recently, entirely different approaches towards gene augmentation are being developed which are based on highly specific modulation of RNA-based regulatory networks [244,245]. The molecular details of these approaches for RNA-targeted gene activation and their potential for clinical translation are discussed by Khorkova et al. [244].

(D) *Genome editing approaches*: These technologies, such as those using CRISPR-Cas9, have proven powerful in stem cells; however, grave challenges remain, such as low rates of homology-directed repair in differentiated somatic cells (e.g., cardiomyocytes, neurons) and risk for off-target effects encompassing the germline.

Despite complex biotechnological challenges, current lack of efficient therapies for multiple severe and abundant diseases is clear evidence for the need to proceed beyond current options. Reassuringly, a remarkable number of pioneering clinical trials have proven technical and clinical feasibility of nucleic acid therapeutic approaches for important cardiovascular [108,112,113,114,115,116,117,118,119,120,121,122,123,124,125,126,127,128,129,130,131,132,133,134,135] and hematological [236,237,238,239,240,241,242] diseases. These are truly fundamental achievements compared to the situation one decade ago.

The versatility of therapeutic non-coding RNA structures will certainly continue to expand the repertoire of our therapeutic tools, with majors leaps to be expected once critical technological issues are solved (Figure 2, Figure 5 and Figure 6). Fundamentally different from DNA, RNAs are carrying information not only in their linear sequences of nucleotides (primary structure), but local nucleotide pairing creates secondary structures, e.g., hairpins, and interactions among distantly located sequences create tertiary structures. In fact, this structural versatility needs to be considered for RNAs as therapeutic tools as well as targets. The plethora of RNA types, sequences, and structures created by evolution is a treasure trove of potential therapeutic tools and targets.

The strategies outlined above involve the use of “informational” drugs designed based on the sequences of specific targets. Additionally, extensive research in the field of RNA-targeted small molecules [234,235,244] complements these approaches and holds promise for enhancing drug delivery to traditionally challenging target tissues, such as the CNS. The current framework primarily employs small molecules as modulators of mRNA splicing and translation. However, it has the potential to expand to target various classes of ncRNAs and encompass both RNA inhibitors and activators. Initial strides have been taken to identify small molecule inhibitors of miRNAs that interfere with miRNA biogenesis [246,247]. Recent work has also out-lined a high-throughput screening strategy for identifying small molecule miRNA modulators using phenotypic expression-based profiling [161,233]. Furthermore, numerous opportunities exist for the targeting of highly structured lncRNAs once their functional domains become better investigated, as reviewed elsewhere [234].

## 11. State-of-the-Art in Nucleic Acid-Based Therapeutics and Their Molecular and Cell-Biological Foundations

It is important to note that nucleic acid-based approaches are currently the only therapeutic tools capable to address a multitude of therapeutic targets with proven key impact upon disease pathogenesis, but without any traditional pharmacological options (small molecule drugs, antibodies). Two decades of intense worldwide efforts to develop the novel approaches towards clinical utility have led to remarkable progress for a number of important diseases and in a few cases their entry into clinical practice. Whereas the liver and hepatocytes emerged as rather easily accessible target for RNAi or ASO strategies, once breakthrough molecular design and delivery discoveries (Figure 5) were made, efficient and selective targeting of many other organs and, in particular, proper targeting of specific cell population therein [248,249,250,251,252] has not yet been successfully applied in clinical trials.

While for many cell types and tissues there is currently still no clinically established targeting system available, this was also the case for liver/hepatocyte targeting just one decade ago. This problem is meanwhile solved through elegant molecular drug design and delivery systems. Considering the putative novel human-specific therapeutic targets discussed above, pathogenic involvement and possible molecular pathomechanisms have already been well documented in cell cultures, organoids, and animal models. Since several of these regard most severe and abundant diseases, this should be considered a strong incentive to solve the remaining methodological hurdles.

Promising progress has recently been made employing direct intrathecal delivery of chemically engineered siRNAs and ASOs to the CNS, enabling long-term target modulation [253,254]. These breakthrough molecular design studies may possibly, in the long run, have similar translational impact as those which have previously led to the successful liver-targeting trials reviewed above. One study engineered a peculiar divalent siRNA chemical scaffold, which enabled potent modulation of gene expression throughout the CNS sustained over six months upon a single intrathecal injection [253]. Another study achieved allele-specific gene silencing in Huntington’s disease models when using chemically engineered siRNAs [254].

An alternative approach to nonviral, chemically synthesized, delivery systems for protein augmentation or RNAi-based target gene silencing is a broad spectrum of recombinant virus-based vectors (Figure 6). Due to the inherently grossly different organ and cell tropisms of the respective basic viruses, their broadly variable stability in the target cells, as well as their potential to induce systemic and local immune responses, selection of a suitable viral vector needs to consider multiple details of the attempted therapeutic strategy [93,94,248,255,256,257,258,259,260,261,262]. These include the desired duration of target modulation, the vector’s capacity to cross relevant anatomical, vascular, and cellular barriers, and preferably even target cell-specific vector–cellular surface receptor interaction to achieve selectivity and avoid off-target side effects.

## 12. Unsolved Challenges and Novel Therapeutic Approaches Guided by Mechanistic Insights at the Molecular and Cell Biological Level

*Molecular and cellular basis of liver targeting*: Similar to the situation about a decade ago, when efficient and hepatocyte-selective *in vivo* delivery systems (Figure 3 and Figure 5) for experimentally already well established targets in the liver were still unavailable, brain or even brain cell-specific drug delivery is still in early infancy today. While hepatocyte-specific genetic drug delivery is clinically applied, for none of a range of brain-targeting approaches based on synthetic nanoparticles [189,191,192,263,264,265,266,267,268,269,270,271], or involving rabies virus proteins [272,273] or vectors [274,275,276,277,278,279,280], the key therapeutic efficacy requirements (Figure 5) have been established so far. Nonetheless, they are most useful for experimental pathogenetic research purposes already.

*Challenges of brain targeting*: Targeted and safe delivery of any nucleic acid-based (siRNA, ASO) drug to specific regions of the brain appears to be a far greater challenge than liver targeting or ex vivo blood stem cell modulation (Figure 7). A remarkable spectrum of brain-targeting approaches encompasses synthetic nanoparticles [189,191,192,263,264,265,266,267,268,269,270,271], rabies virus proteins [272,273], or vectors [274,275,276,277,278,279], yet none of these are established with respect to key efficacy requirements (Figure 5). Nonetheless, they are already most useful for pathogenetic research. Recombinant AAV vectors are currently encountering particular interest for brain-directed therapies due to favorable inherent or engineered properties, as outlined below.

*First experimental steps*: A number of first experimental therapeutic studies in mice (Section 3) holds promise for further developments with translational potential employing more sophisticated and safe drug delivery systems. These will need to have well-established molecular and cellular mechanisms of action, including carrier ligand–cell surface receptor [184,185,186,187,188,189,281] interactions, as well as clearly characterized *in vivo* behavior [191,192,193]. The latter encompasses possible crossing of the blood–brain barrier via transcytosis [282,283,284,285,286], as well as local drug delivery through stereotactic approaches. Furthermore, non-invasive monitoring of therapeutic drug function in patients is highly desirable. For therapeutic proteins this may be enabled by a magnetic resonance imaging (MRI) approach [287,288,289] for real-time reporting of the gene therapy product *in vivo* by use of an MRI probe that is activated in the presence of therapeutic protein expression.

*Advanced AAV-based vector systems*: Currently, AAV-based vectors are encountering high interest for brain-targeted therapies [248,290,291,292]. In other fields, e.g., hematology or cardiovascular medicine, genetically engineered and surface-modified (pseudotyped) versions of this vector have been extensively studied previously for therapeutic gene augmentation [258,293] and RNA interference (RNAi) mediated gene silencing [93]. AAVs are considered as vectors of choice for many nervous system targets due to desirable safety profile, extensive basic science, and clinical experience from other fields including clinical trials, stable transgene expression in post-mitotic cells, and neuronal tropism. Low immunogenicity of AAVs is a further critically important aspect, and the recently developed system of extracellular vesicle-encapsulated AAVs (EV-AAVs) enables efficient gene transfer even in the presence of pre-existing AAV-neutralizing antibodies in patients [258,294,295,296].

*Anatomical barriers against nanoparticle or vector-based therapeutics:* One study [291] reported that a particular serotype, AAV9, is capable to cross the blood–brain barrier (BBB) [286,297], raising the possibility of intravascular administration as a non-invasive delivery route to achieve widespread CNS gene expression (Figure 7). Crossing of the blood–brain barrier appears to occur via transcytosis [282,283,284,285]. Notably, this same AAV serotype is also capable to enter the myocardium across the tight cardiovascular endothelium (impermeable for other AAV serotypes) and has previously been successfully employed for cardiac gene transfer [258,293] and the first demonstration of cardiac RNA interference (RNAi) therapy [93].

*Differential tropism and cell type selectivity*: So far, however, very little is known about differential tropism of the currently available AAV pseudotypes for different brain cell types of specific therapeutic interest [298,299]. From previous studies, it is well known that genetic engineering of the vector surface i.e., pseudotyping has, in principle, the capacity to alter tropism in a desired direction [248,250,291,292,300,301]. However, recent high-throughput methods have identified the host proteins essential for vector attachment and internalization more comprehensively, and subsequent molecular studies including cryogenic electron microscopy (cryo-EM) have revealed unanticipated complexity and serotype specificity of the cellular vector entry process [302]. Theoretical predictability of the *in vivo* effects of vector modifications is therefore limited and extensive experimental validation essential [303].

Pillay et al. [304] used an unbiased genetic screen to identify proteins essential for AAV infection and identified a previously uncharacterized type I transmembrane protein, KIAA0319L, which they named AAV receptor (AAVR). They characterized AAVR as capable of rapid endocytosis and trafficking to the trans-Golgi network. AAVR was a critical host factor for all tested AAV serotypes and AAVR^−/−^ mice were resistant to AAV infection, establishing AAVR as a universal receptor for involved in AAV infection. GPR108, a member of the G protein-coupled receptor superfamily, was subsequently identified as another highly conserved AAV entry factor [305]. Among 20 divergent AAVs across all AAV clades, only AAV5 transduction was unaffected in the GPR108 knockout (KO). Thus, this study identified the second of two AAV entry factors conserved between mice and humans and relevant in vitro and *in vivo*.

Starting from the AAV9 serotype holding promise for trans-BBB therapy, a recombinant AAV-PHP.eB was engineered by insertion of a 7-amino acid peptide and point mutations of neighboring residues into the AAV9 capsid, thereby enhancing potency in the central nervous system [306]. Consideration will be required for translation beyond mouse models, however, because the CNS transduction benefits of AAV-PHP.eB over AAV9 are dependent on administration route and mouse strain [307]. Specifically, AAV-PHP.eB produced higher CNS transduction than AAV9 after intravenous injection, but only in C57BL/6J and not in B6C3 mice. Another study [308] found mutation of certain tyrosine (Tyr) residues on the AAV2 capsid enhanced neuronal transduction in striatum and hippocampus, while analogous Tyr substitutions on AAV5 and AAV8 did not, emphasizing the unforeseeable impact of any modifications with regard to both targeting efficacy and specificity. Similar challenges with regard to clinical translation, generated by species differences, have been extensively investigated with regard to another “hard target” for nucleic acid therapeutics, i.e., the myocardium [8,93,258,294,309,310,311].

One approach towards selective targeting of *specific* brain cell types are engineered AAVs deficient in heparan-sulfate proteoglycans (HSPGs) attachment, but instead recognizing the glutamate receptor 4 (GluA4) through a displayed GluA4-specific DARPin (designed ankyrin repeat protein). When injected into mouse brain, >90% of the transduced cells were interneurons. The DARPin mediated selective vector *attachment* to GluA4-positive cells, while actual transgene *delivery* still required expression of AAVR [312].

Another highly innovative strategy employs membrane protein-specific nanobodies inserted into a surface loop of the VP1 capsid protein of AAV2. Nanobodies are single immunoglobulin variable domains of heavy chain antibodies naturally occurring in camelids [313]. Nanobodies specific for different membrane proteins dramatically enhanced the transduction of specific target cells by recombinant AAV2. Nanobody-VP1 fusion was incorporated into AAV1, AAV8, and AAV9 and effectively re-directed the target specificity of these AAV serotypes, too.

*Transgene expression control:* Beyond stability in the blood circulation and capability to cross the blood–brain barrier, it is also desirable to equip the vector with a promoter providing long-term transgene expression in the brain. Maturana et al. [314] have identified and characterized small alphaherpesvirus latency-associated promoters (LAPs) which enabled stable, pan-neuronal transgene transcription and translation from AAV-LAPs in the CNS for 6 months. Thus, these LAPs are suitable candidates for AAV-based CNS gene therapies requiring chronic transgene expression after one-time viral-vector administration.

## 13. Summary and Outlook

Figure 7 highlights the fact that liver-targeted genetic therapies (both for protein augmentation or gene silencing) are established for several important human metabolic and monogenic disease, since the key problem of proper targeting and stability upon systemic injection has been solved. In principle, genetic modulation of immune cells might expand the currently available arsenal of small molecule drugs, antibodies, and therapeutic cells. Except for ex vivo transduction of progenitor cells, however, immune cell subtype-specific transduction for the treatment of inborn haematological or immune diseases is still immature. The ultimate goal in neurology would be crossing of the intact blood–brain barrier (e.g., transcytosis), followed by spontaneous vascular egress and selective but efficient entry of the therapeutic nucleic acid into the target cell type where pathogenesis occurs.

It must be kept in mind that each new nucleic acid-based drug and therapeutic target may raise specific and previously unexpected issues, the most critical ones regarding safety (immune activation, off-target effects at the cellular level, systemic mis-targeting and accumulation, delayed-onset safe effects). It should be emphasized that each individual, e.g., siRNA or ASO drug formulation, may have a specific side-effect, or particularly high efficacy, which is not ‘group-specific’ i.e., not common to the siRNA or ASO class of drugs in general. Seemingly minute molecular details of a siRNA, ASO, or gene therapy drug may determine whether, for example, immune reactions or thrombopenia will be triggered by this individual compound. High vigilance therefore needs to be focused upon each individual drug from the field of novel nucleic acid-based and epigenetic therapies.

## 14. Conclusions

There is an expanding spectrum of diseases for which nucleic acid-based therapeutics addressing fundamentally new therapeutic targets are envisaged or under development. Extensive genetic, experimental, and clinical work has highlighted important new therapeutic targets possible causal or significantly contributing to development of these grave diseases.

Synergistic with the most remarkable recent progress regarding delivery, efficacy, and safety of nucleic acid-based therapies, past and ongoing large-scale exploration of the noncoding genome for human-specific therapeutic targets is encouraging to proceed with the development and clinical evaluation of such new therapeutic pathways.

## Figures and Tables

**Figure 1 cells-12-02660-f001:**
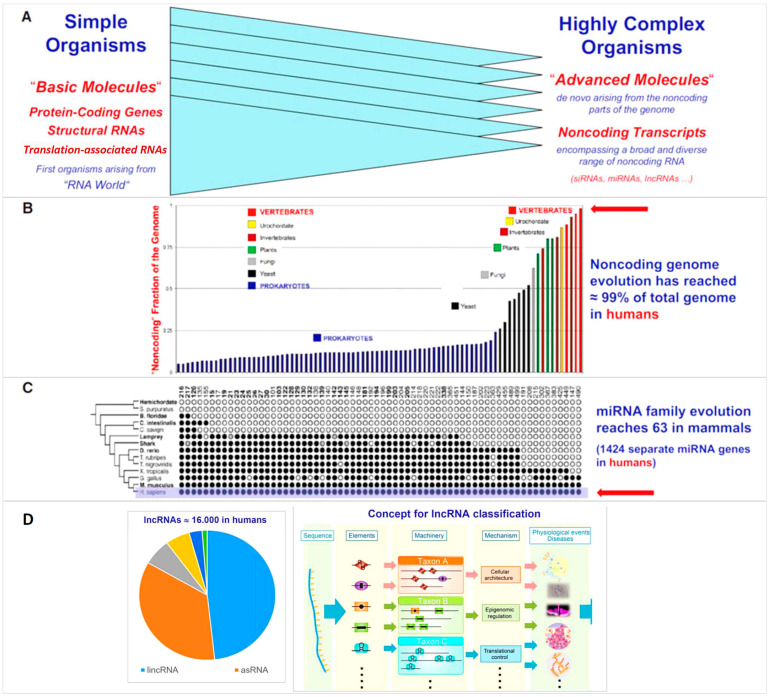
Evolutionary expansion of the noncoding genome in higher organisms. (**A**) The basic molecules of life already found in the earliest and most simple organisms are increasingly supplemented, during the evolution to complex species, by molecules needed for correct embryonic development and homeostatic stability of their morphology and functions. (**B**) Whereas the number of protein-coding genes remains similar from simple to complex species, it is the noncoding part of the genome that increases dramatically with morphological complexity to >98% in humans. (**C**) Few types of noncoding RNAs arising from the noncoding genome have been phylogenetically mapped in depth. Thus, investigation of microRNA (miRNA) family evolution revealed impressive increases with the advent of vertebrates, and ancient miRNAs families can well be distinguished from those more recently arising. (**D**) No definitive classification of the huge number of lncRNAs has been established so far. Several basic elements suitable as components for classification are shown, encompassing sequence elements, conserved structural motifs, mechanisms of action, and physiological or disease processes in which the respective lncRNAs are involved (Modified from Poller et al. 2013 [9] by permission of Circ. Res.).

**Figure 2 cells-12-02660-f002:**
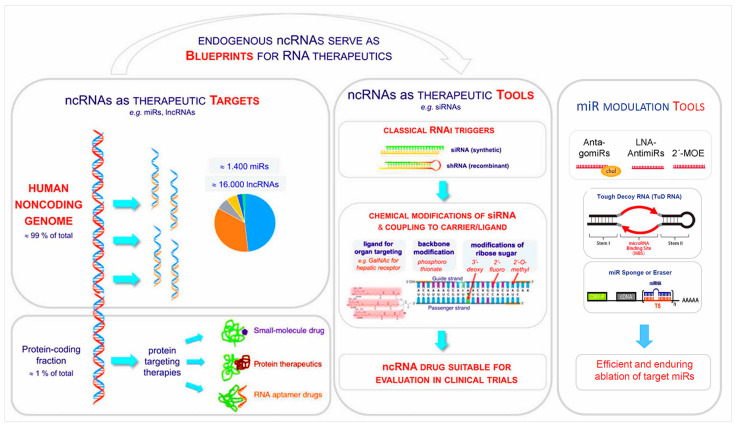
Endogenous non-coding RNAs as blueprints for RNA therapeutics. Non-coding RNAs may be addressed as therapeutic targets, but an increasing spectrum of endogenous ncRNAs (e.g., siRNAs) are also employed as blueprints for the development of novel therapeutic tools. The spectrum of possible therapeutic targets has vastly expanded beyond proteins, but likewise the therapeutic ‘toolbox’. One current topic is therapeutic RNA interference triggers (siRNAs) originally developed from endogenous siRNAs as blueprints, and made clinically applicable based on sophisticated chemical modifications and coupling to carriers/ligands for tissue targeting. Appreciation of the profound pathogenic impact of diverse small and long ncRNAs has inspired the development of multiple other therapeutic tools engaging these ncRNAs. The tools may be engineered nucleic acids themselves, acting through sequence homologies, or “classical” small molecule drugs designed to interact with e.g., conserved 3D *structural* motifs in lncRNAs which are not necessarily dependent on strict RNA *sequence* conservation.

**Figure 3 cells-12-02660-f003:**
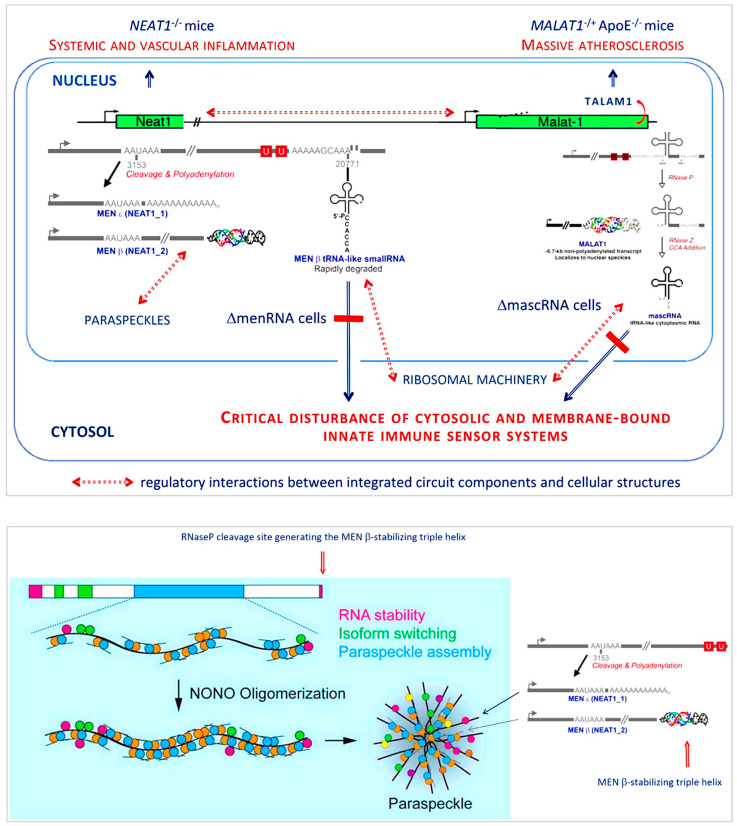
Multi-level functional integration of extended regions of the human genome, above and beyond individual noncoding RNAs. The *NEAT1–MALAT1* genomic region encodes a biologically integrated circuit controlling innate immune sensing and cell–cell interactions. From an evolutionary perspective, the NEAT1–MALAT1 genomic region appears as a highly integrated RNA processing circuitry critically contributing to immune homeostasis. Its components MEN-β, MEN-ε, menRNA, MALAT1, TALAM1, and mascRNA are obviously set for well-balanced interactions with each other. Genetic ablation of any element therefore leads to major dysfunction. Beyond prior work in NEAT1 and MALAT1 knockout mice, a recent cell biological study identified menRNA and mascRNA as novel components of innate immunity with deep impact upon cytokine regulation, immune cell–endothelium interactions, angiogenesis, and macrophage formation and functions. These tRNA-like transcripts appear to be prototypes of a class of ncRNAs distinct from other small transcripts (miRNAs, siRNAs) by biosynthetic pathway (enzymatic excision from lncRNAs) and intracellular kinetics, suggesting a novel link for the apparent relevance of the NEAT1–MALAT1 cluster in cardiovascular and neoplastic diseases. For the long primary transcripts of NEAT1, a function of general cell-biological interest has been identified. They are critical for the formation of paraspeckles which are involved in multiple cellular functions, and possibly also in the broader context of micellization and the formation of biomolecular condensates essential for proper subcellular and nuclear compartmentalization. Obviously, molecules involved in these fundamental processes may deeply impact upon various cellular functions in a context-dependent manner, so that their observed association with diverse diseases is therefore not entirely unexpected. Overall, the NEAT1–MALAT1 genomic region may serve as paradigm of a biological integrated circuit fine-tuning multiple cellular processes covering innate immune sensing and cell–cell interactions. (Modified from Poller et al. 2023 [11] by permission from J. Clin. Med.).

**Figure 4 cells-12-02660-f004:**
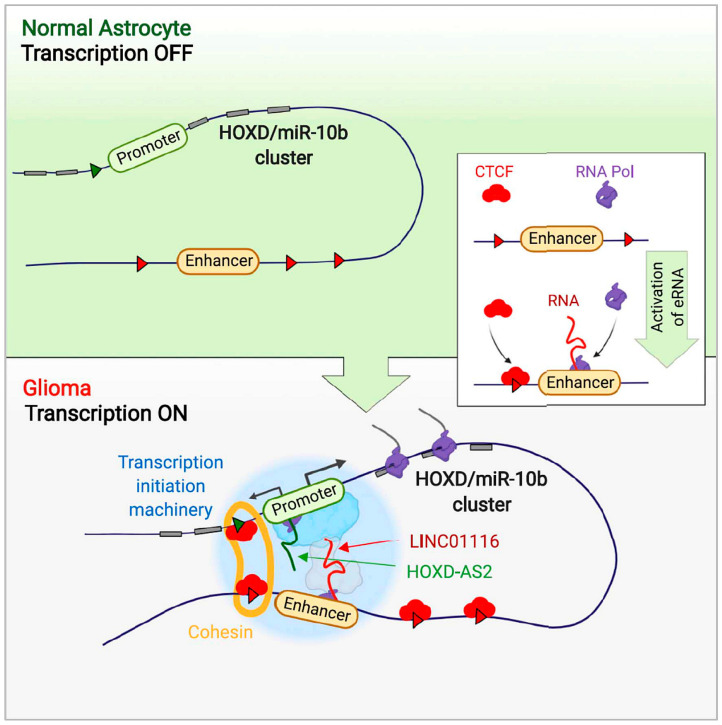
Promoter and enhancer RNAs regulate chromatin reorganization and activation of miR-10b/HOXD locus, and neoplastic transformation in glioma. miR-10b is silenced in normal neuroglial cells of the brain but commonly activated in glioma, where it assumes an essential tumor-promoting role. The entire miR-10b-hosting HOXD locus is activated in glioma via the cis-acting mechanism involving 3D chromatin reorganization and CTCF-cohesin-mediated looping. This mechanism requires two interacting lncRNAs, HOXD-AS2 and LINC01116, one associated with HOXD3/HOXD4/miR-10b promoter and another with the remote enhancer. Knockdown of either lncRNA in glioma cells alters CTCF and cohesin binding, abolishes chromatin looping, inhibits the expression of all genes within HOXD locus, and leads to glioma cell death. Conversely, in cortical astrocytes, enhancer activation is sufficient for HOXD/miR-10b locus reorganization, gene derepression, and neoplastic cell transformation. LINC01116 RNA is essential for this process. Our results demonstrate the interplay of two lncRNAs in the chromatin folding and concordant regulation of miR-10b and multiple HOXD genes normally silenced in astrocytes and triggering the neoplastic glial transformation. (Modified from Deforzh et al. 2022 [88] by permission from Mol. Cell).

**Figure 5 cells-12-02660-f005:**
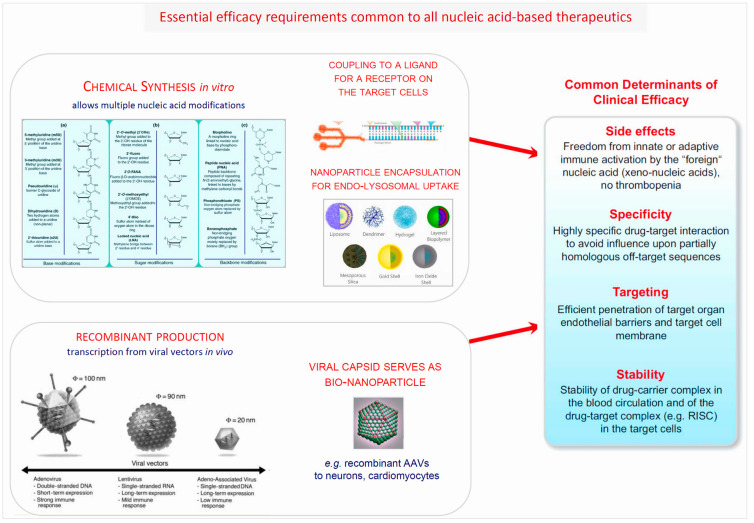
The fundamental clinical efficacy determinants of ASO and siRNA therapeutics. Despite broad diversity of the new nucleic acid-based therapeutic principles and tools, they share key common determinants of clinical efficacy which are critical for possible translational success and need to be closely monitored in any clinical trial.

**Figure 6 cells-12-02660-f006:**
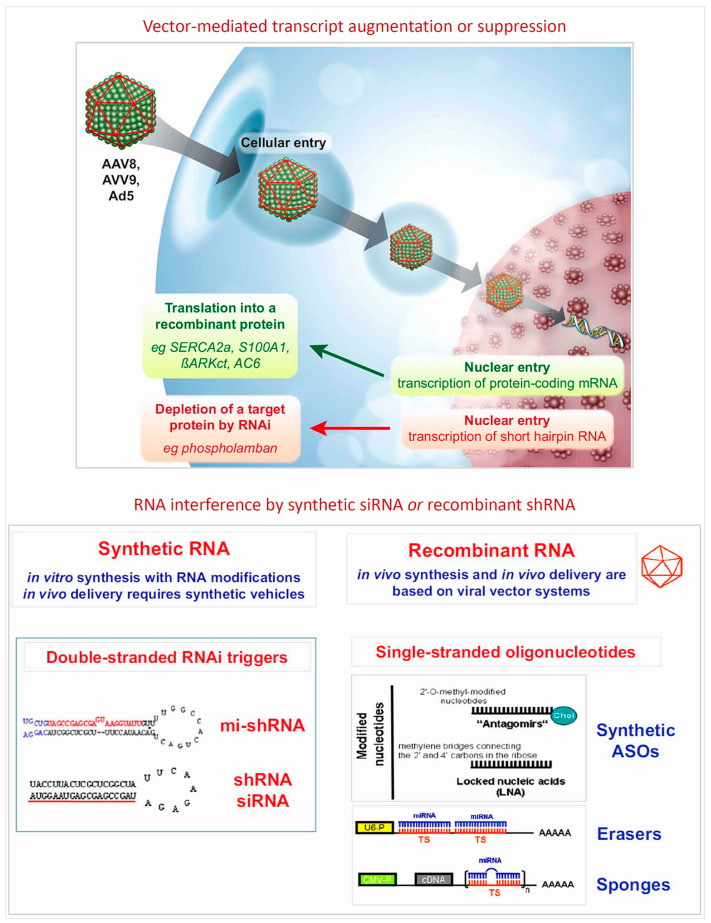
Vector-based genetic therapies for protein augmentation or RNAi-mediated target RNA depletion. The ‘classical’ approach of gene transfer for protein *augmentation* or (in the case of monogenic disorders) protein *substitution* recently gained clinical impact in the hemophilia field where the missing coagulation factor genes could be successfully and durably transferred to the liver using AAV vectors. In the cardiovascular field, cardiac-targeted gene augmentation (SERCA2a) or ablation (phospholamban) therapies were successful in animal models, but this could not yet be translated to the clinical arena due to as yet insufficient gene transfer efficacy in patients. The opposite approach is post-transcriptional silencing of genes involved in disease pathogenesis. Complementary to chemically synthesized base- and backbone-modified ASOs or siRNAs (Figure 6), silencing of any protein-coding or noncoding transcript may be achieved by viral vector-based RNA interference (RNAi). Two fundamentally distinct approaches (lower panel) use synthetic siRNAs, or recombinant shRNAs continuously produced from viral vectors. RNA is inherently unstable and must be modified to achieve sufficient biostability, and delivered via synthetic carriers, to become therapeutically useful. Viral vectors, which may be organ-targeted and regulatable, may circumvent targeting issues by their inherent biological properties, and the RNA stability problem by continuous synthesis in the host cells. Apart from these differences, the same characteristics will be considered when the therapeutic potential of synthetic or recombinant RNA drugs is assessed. AAV indicates adeno-associated virus; ASO, antisense oligonucleotide; LNA, locked nucleic acid; shRNA, short hairpin RNA; siRNA, short interfering RNA; TS, target site. (Modified from Poller et al. 2013 [9] by permission from Circ. Res.).

**Figure 7 cells-12-02660-f007:**
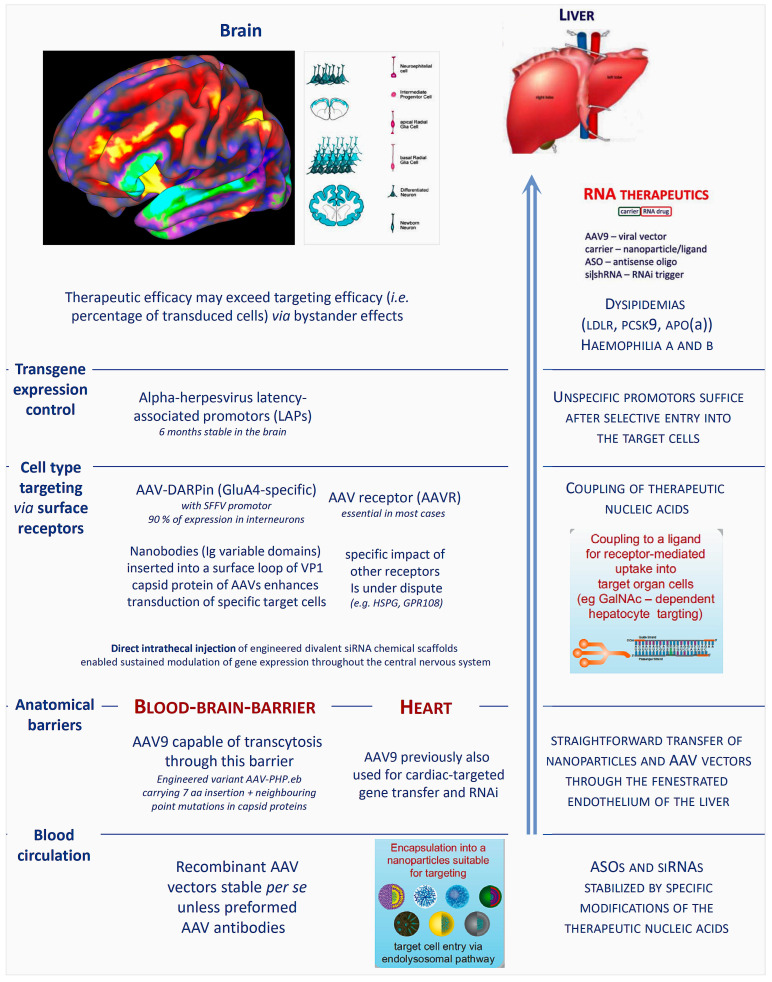
Summary and outlook. On the right side, this figure aims to summarize- the molecular and cellular basis of liver targeting of nucleic acid-based drugs, which is already being applied clinically for several disorders. Here, drug delivery is facilitated by the fenestrated endothelium of the liver and the availability of a safe and efficient hepatocyte-specific ligand-receptor system. The left side outlines the multiple challenges arising once targeting to other organs is attempted, focusing upon brain delivery. Targeted and safe delivery of any nucleic acid-based (siRNA, ASO) drug to specific regions of the brain appears as far greater challenge than liver targeting or ex vivo blood stem cell modulation. A remarkable spectrum of brain-targeting approaches encompasses synthetic nanoparticles and viral vector, yet so far, none of these are established with respect to key efficacy requirements (Figure 5). AAV vectors encounter high interest for brain-targeted therapies since genetically engineered and surface-modified (pseudotyped) versions of this vector have been extensively studied in other fields of medicine (e.g., hematology, cardiovascular medicine). The blood-brain-barrier (BBB) constitutes a particularly challenging anatomical barrier against nanoparticle or vector based drug delivery. Remarkably, the serotype AAV9 is capable to cross the BBB under certain conditions, raising the possibility of intravascular administration as a non-invasive delivery route of nucleic acid-based drugs to the CNS. Noteably, this same AAV serotype is also able to enter the myocardium across the tight cardiovascular endothelium (impermeable for other AAV serotypes) and was previously employed for cardiac-targeted gene transfer and RNA interference therapy. Regarding the next step of delivery, little is known about differential tropism of currently available AAV variants for distinct brain cell types of specific therapeutic interest. Recent high-throughput screens have identified host proteins essential for AAV delivery in a comprehensive manner and revealed unanticipated complexity and serotype specificity of the entry process. Theoretical predictability of any in vivo effects of vector modifications is therefore limited and experimental validation essential. The figure depicts recent experimental approaches to improve BBB passage and brain cell type-specific delivery. Starting from AAV9 holding promise for trans-BBB therapy, AAV-PHP.eB was engineered by insertion of a 7-amino acid peptide and point mutations of neighboring residues into the AAV9 capsid and enhanced CNS delivery in mice only under certain conditions. Similar challenges with regard to clinical translation, generated by species differences, have been extensively investigated before for another “hard target”, i.e., the heart. Instead of recognizing the glutamate receptor GluA4 through a displayed GluA4-specific DARPin, AAVs deficient in HSPGs attachment resulted in preferential >90% transduction of interneurons. Another highly innovative strategy employs membrane protein-specific nanobodies inserted into a surface loop of the VP1 capsid protein of AAVs. Nanobody-VP1 fusion was applied to AAV1, AAV2, AAV8, and AAV9 and effectively re-directed the target specificity of all these AAV serotypes. Beyond stability in the blood circulation and capability to cross the blood–brain barrier, transgene expression stability or even control is also desirable. Alphaherpesvirus latency-associated promoters (LAPs) enabled stable, pan-neuronal transgene transcription and translation from AAV-LAPs in the CNS for 6 months. Thus, these LAPs are suitable candidates for AAV-based CNS gene therapies requiring chronic transgene expression after one-time viral-vector administration.

## Data Availability

Data are contained within the article.

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
