# Peer review of "Exploration of the Noncoding Genome for Human-Specific Therapeutic Targets—Recent Insights at Molecular and Cellular Level"

_cells, 2023, doi:10.3390/cells12222660_

Round 1
Reviewer 1 Report
Comments and Suggestions for Authors
The manuscript is a review on the therapeutic possibilities of non-coding RNAs. The text is well written. The article begins with a general description of the functions of non-coding RNAs, then goes on to discuss some non-coding RNAs, and then describes a number of diseases associated with the expression of non-coding RNAs. It would be good to classify the functions of non-coding RNAs in the article - microRNA, siRNA, antisense RNA, lncRNA etc. Among the non-coding RNAs are RNAs associated with chromatin modifications. In further description, it is good to indicate the type of RNA. RNAs associated with chromatin modifications are not described when describing therapeutic prospects. Are there really no such prospects?
In the analysis, the authors implicitly assume that humans are the most complex organism (Figure 1). Is this the case? Maybe the large number of non-coding RNAs found in humans is a consequence of humans being the most studied organism. It is possible that an equally deep study of, say, octopuses or amoebae would also show a high proportion of non-coding RNAs. Moreover, infusoria have a very large number of non-coding guide RNAs, not to mention other types of RNAs.
Lines 224-232: It should be noted that CHROMR and CHROME are antisense RNAs.
Line 249: The statement "From the discovery that about 98-99% of the human genome do not encode proteins, but instead generate a broad spectrum of ncRNAs..." is rather controversial, since the functionality of transcription of almost the entire genome is not at all obvious, and may be either an experimental artifact or biological transcriptional noise. Transcription of almost the entire genome involves transcription of repeats, pseudogenes, retroviruses, and so on. At the level of mammals and even primates, the level of conservation of the non-coding genome is not so great as to suggest selection pressure, and hence its functionality. A more precise formulation should be used here.
Line 393: abbreviation ADRD not defined.
Line 480: abbreviation ASO not defined.
The quality of the drawings needs to be fundamentally improved. Even on zooming it is not possible to read the texts in the figures.
Reviewer 2 Report
Comments and Suggestions for Authors
In this very complete review authors describe usage of noncoding genome for for therapeutic Targets –
I two comments:
the abstract is quite completely identical to the introduction. I asked the author whether is possible to change it. Moreover, I do not appreciate too much the presence of q uestion into an introduction or abstract section. Is it possible to riformulate them in another way?
Although figures into the paper have permission, it should be nice whether authors make them as original.
